# Vaccinating Adolescents and Children Significantly Reduces COVID-19 Morbidity and Mortality across All Ages: A Population-Based Modeling Study Using the UK as an Example

**DOI:** 10.3390/vaccines9101180

**Published:** 2021-10-15

**Authors:** Tinevimbo Shiri, Marc Evans, Carla A. Talarico, Angharad R. Morgan, Maaz Mussad, Philip O. Buck, Phil McEwan, William David Strain

**Affiliations:** 1Health Economics and Outcomes Research Ltd., Cardiff CF23 8RB, UK; tinevimbo.shiri@heor.co.uk (T.S.); angharad.morgan@heor.co.uk (A.R.M.); phil.mcewan@heor.co.uk (P.M.); 2Diabetes Resource Centre, University Hospital Llandough, Cardiff CF64 2XX, UK; marclyndon1@hotmail.com; 3Moderna Inc., Cambridge, MA 02139, USA; Carla.Talarico@modernatx.com (C.A.T.); Philip.Buck@modernatx.com (P.O.B.); 4Ashfield Healthcare on Behalf of Moderna, Ashby-de-la-Zouch LE65 1HW, UK; Maaz.Mussad@modernatx.com; 5Diabetes and Vascular Research Centre, University of Exeter Medical School, Exeter EX1 2HZ, UK; 6The Academic Department of Healthcare for Older Adults, Royal Devon and Exeter Hospital, Exeter EX2 5DW, UK

**Keywords:** adolescent, coronavirus, disease transmission model, COVID-19, SARS-CoV-2, vaccination

## Abstract

Debate persists around the risk–benefit balance of vaccinating adolescents and children against COVID-19. Central to this debate is quantifying the contribution of adolescents and children to the transmission of SARS-CoV-2, and the potential impact of vaccinating these age groups. In this study, we present a novel SEIR mathematical disease transmission model that quantifies the impact of different vaccination strategies on population-level SARS-CoV-2 infections and clinical outcomes. The model employs both age- and time-dependent social mixing patterns to capture the impact of changes in restrictions. The model was used to assess the impact of vaccinating adolescents and children on the natural history of the COVID-19 pandemic across all age groups, using the UK as an example. The base case model demonstrates significant increases in COVID-19 disease burden in the UK following a relaxation of restrictions, if vaccines are limited to those ≥18 years and vulnerable adolescents (≥12 years). Including adolescents and children in the vaccination program could reduce overall COVID-related mortality by 57%, and reduce cases of long COVID by 75%. This study demonstrates that vaccinating adolescents and children has the potential to play a vital role in reducing SARS-CoV-2 infections, and subsequent COVID-19 morbidity and mortality, across all ages. Our results have major global public health implications and provide valuable information to inform a potential pandemic exit strategy.

## 1. Introduction

The ongoing COVID-19 pandemic, caused by severe acute respiratory syndrome coronavirus 2 (SARS-CoV-2), has resulted in more than 193 million confirmed cases of COVID-19 and more than 4 million deaths globally, as of 26 July 2021 [1]. It has been estimated that at least one in five people infected with COVID-19 experience persistent ill health following the acute phase of infection. This condition is typically referred to as long COVID; it includes both ongoing symptomatic COVID-19, with signs and symptoms continuing for 4 to 12 weeks after acute COVID-19, as well as post-COVID-19 syndrome, with signs and symptoms that develop during or after acute COVID-19 continuing for more than 12 weeks, and which cannot be explained by an alternative diagnosis [2].

Vaccinations against SARS-CoV-2 reduce serious illness and transmission, thereby limiting the health and socioeconomic impacts of the pandemic. The global rollout of vaccines to adults is proceeding apace, with the World Health Organization (WHO) reporting that 3.6 billion vaccine doses have been administered as of 25 July 2021 [1]. The success of vaccination programs depends upon multiple factors, including vaccine efficacy, the speed of vaccine rollout and uptake rates, the emergence of new variants, patterns of social mixing and, critically, population vaccination strategies. In countries where an increasing percentage of adults are fully vaccinated, but where children and adolescents are not, debate persists as to the population relevance of the high prevalence of SARS-CoV-2 in children and adolescents, and the role of vaccinating younger age groups in controlling the pandemic. In the UK, vaccination of those over the age of 12 years is, as of 30 July 2021, limited to vulnerable individuals and those in households with immunocompromised members; mass vaccination of children and adolescents is not currently planned [3].

Currently, the AstraZeneca (Oxford), BioNTech/Pfizer, and Moderna COVID-19 vaccines are available across the UK, all of which provide effective protection against coronavirus infection, and even better protection against serious illness and disease. There has been a small number of cases of severe side-effects and deaths among those receiving the AstraZeneca vaccine. However, for people aged 30 years or over and those with other health conditions, the benefits of being vaccinated outweigh any risk associated with this vaccine. For people aged under 30 years and without other health conditions, it is advised that they receive an mRNA vaccine from Pfizer/BioNTech or Moderna as an alternative to this vaccine. The efficacy and safety of these mRNA vaccines in those under 18 years of age has recently been evaluated, with both Pfizer/BioNTech’s BNT162b2 vaccine and Moderna’s mRNA-1273 vaccine demonstrating 100% vaccine efficacy in phase 3 trials of adolescents, and with only mild side effects, generally consistent with those observed in individuals ≥18 years [4,5]. Based on these results, Pfizer/BioNTech’s vaccine has been approved for adolescents ≥12 years in the US, as well as in European Union member states and in the UK [6,7,8]. Moreover, the European Medicines Agency’s (EMA) Committee for Medicinal Products for Human Use (CHMP) has recently adopted a positive opinion, recommending marketing authorization for Moderna’s vaccine to include adolescents ≥12 years [9], with other nations following suit.

Given uncertainty regarding the potential benefit of vaccinating adolescents and children, coupled with the economic pressure to relax population restrictions globally, there is a pressing need to evaluate the public health impact of vaccinating individuals <18 years old. In recent years, our understanding of infectious disease epidemiology has been greatly increased through mathematical models, which have generated quantitative information to inform public health planning and infectious disease control [10,11,12]. Such mathematical models have become a central tool in the control of COVID-19 in many countries; they can provide short- and long-term forecasts regarding the future direction of the COVID-19 pandemic, and provide a framework for evaluating the potential impact of different vaccination strategies. The most relevant mathematical models relating to the spread of a pandemic are compartmental models, with the population divided into different population groups based on their infection status (e.g., susceptible, exposed, infectious, recovered), and with assumptions being made about the rate of transfer from one group to another. Here, we use a novel Susceptible-Exposed-Infectious-Recovered (SEIR) model to quantify the impact of COVID-19 variants on disease and transmission, using the UK as an example due to the country’s high COVID-19 vaccination rates in adults, the high prevalence of the delta variant, and the lifting of all COVID-19 restrictions as of 19 July 2021. We evaluated the impact of returning to pre-pandemic levels of social mixing in the UK from 19 July 2021 on future COVID-19 pandemic dynamics. In addition, we assessed the population level impact of extending the vaccination program to all adolescents, as well as to all adolescents and children, on SARS-CoV-2 infection rates, hospitalization, mortality, and rates of long COVID. 

## 2. Methods

### 2.1. Disease Transmission Model Overview

We developed an age-stratified, deterministic SEIR compartmental model of SARS-CoV-2 infections (Figure 1), structured to evaluate transmission within nine incremental age bands of 10 years each, from 0 years old to 79 years old, and an additional age group comprising those 80+ years old.

In the model, the uninfected population originates in the ‘susceptible (S)’ state and progresses through epidemiological stages once infected. Once exposed to the virus, transition from the ‘susceptible’ state to the ‘exposed (E)’ state occurs, where the infection is latent and not yet infectious (the ‘incubation period’). Subsequently, the infected may develop symptoms (although a large proportion will remain asymptomatic) and transition to the ‘infectious (I)’ state. Those within the infectious state may either transition to the ‘recovered (R)’ state once the infection has abated, or develop severe disease, requiring hospitalization (H) and possibly admission to an intensive care unit (ICU). Furthermore, a proportion of infected individuals may develop long COVID (LC). Age-dependent rates are utilized to model the incidence of long COVID, hospitalization, and mortality (either within the community or in hospital). Those critically ill patients may recover, or they may die due to COVID-19.

We used this model to characterize the SARS-CoV-2 infection dynamics from the beginning of the UK epidemic to validate the model’s capacity to extrapolate beyond the present day. The model captures the impact of restrictions, such as lockdown and social distancing, using age-dependent and calendar time-dependent contact matrices. Therefore, the model can predict short-term infection dynamics and assess any policy changes regarding epidemic management. It can further predict the impact if the UK were to expand the vaccination program to include adolescents (≥12 years) and children (≥5 years) using an mRNA vaccine. A full description of the parameters and assumptions in the model are provided in the Appendix A.

### 2.2. Scenarios Evaluated

The Base Case (BC) follows current UK guidance, using vaccine coverage projections for adults ≥ 18 years and adolescents aged ≥12 years at increased risk of serious COVID-19 disease and hospitalization. We assume that this represents approximately 370,000 adolescents of 12–17 years who are classified as vulnerable, and that coverage will be 100% in this vulnerable group, representing an optimistic scenario. 

The Base Case plus Adolescents (BCA) scenario is as per the BC scenario but extended to assume vaccine coverage in 12–17-year-olds is 100%, representing an optimistic scenario (and with alternative scenarios explored in sensitivity analysis). The 2019, mid-year estimates for the UK calculate the population of 12–17-year-olds as 4.5 million (4,482,216).

The Base Case, Adolescents and Children (BCAC) scenario is as per the BCA scenario but extended to assume all children 5–11 years old are vaccinated (approximately an additional 4.1 million (4,149,852) children). 

Appendix A illustrate the vaccine coverage and rate of uptake applied for each scenario: BC, BCA, and BCAC, respectively.

### 2.3. Sensitivity Analysis

For the three scenarios outlined above, we assumed that the effective reproductive number (R_eff_), defined as the number of secondary infections produced by a single infectious individual when the population has both susceptible and non-susceptible (e.g., infectious, immune, vaccinated) individuals, and/or interventions are in place to combat the spread of infection, remains at 1.4 until 31 December 2021. This is based on the assumption that the voluntary use of masks and social distancing indoors is maintained, and that the delta variant remains the dominant strain, with no new variants emerging. Scenarios for R_eff_ 1.7 (the highest seen in the UK during the second wave) and R_eff_ 2.1 (estimated value from the Netherlands two weeks after lifting restrictions) were also explored. In addition, we assessed scenarios when we assumed that maximum vaccine coverage in adolescents and children could be either 50% or 75%.

## 3. Results

Under BC settings, we estimated a further 3.8 million SARS-CoV-2 infections in the UK between July and December 2021. Assuming no change in vaccination strategy or social distancing measures, daily SARS-CoV-2 infections would approach approximately 300,000 per day, with daily hospitalizations and deaths approaching 1400 and 150, respectively (Figure 2).

This would be expected to result in a total of approximately 129,000 additional hospital admissions and 13,000 excess deaths (Table 1). Extending vaccinations to also include adolescents and children would be expected to reduce SARS-CoV-2 infections to approximately 2 million (26% reduction) and 736,000 (64% reduction), respectively, over the same period. Hospital admissions and deaths would decrease to approximately 101,000 (21% reduction) and 10,500 (18% reduction), respectively, (BCA), and to 51,000 (60% reduction) and 5500 (57% reduction), respectively, (BCAC). An additional 80,000 cases of long COVID would be expected to occur between July to December 2021, with the potential to be reduced to 58,000 (27% reduction) and 20,000 (75% reduction) when extending vaccination to adolescents and children, respectively.

With the existing vaccination strategy (BC), approximately 60% of new infections would be in those <18 years, with these individuals accounting for 14% of hospital admissions, 61% of long COVID cases, and 0.2% deaths (Figure 3). While the mortality rate in children <18 years is low (20 cases would be expected under the base case scenario), the BCAC vaccination strategy would be expected to reduce overall COVID-related mortality by 57%.

Figure 4 illustrates the patterns of benefit across age groups associated with extending vaccination to children, with children and adolescents principally benefiting from fewer long COVID cases (approximately 40,000), and older people (>60 years) having fewer deaths (approximately 4000). The pattern of hospitalizations avoided is also noteworthy, with adolescent vaccination contributing to a reduction of approximately 5700 events in the age group, which, in absolute terms, is greater than any single adult age group.

Sensitivity analyses demonstrate that an increasing R_eff_ has a substantial impact upon the burden of SARS-CoV-2 infections. Under the BC scenario, infections would be expected to increase from 2.7 million to 7.4 million and 19.0 million for R_eff_ 1.4, 1.7, and 2.1, respectively. These additional infections would be associated with a corresponding increase in hospitalizations, deaths, and long COVID cases (Table 1). Furthermore, sensitivity analyses also highlight the impact associated with reduced vaccine coverage. Under the BC scenario, infections would be expected to increase from 2.7 million to 3.0 million and 3.1 million for vaccine coverage of 100%, 75%, and 50%, respectively. These additional infections would be associated with a corresponding increase in hospitalizations, deaths, and long COVID cases (Table 1).

## 4. Discussion

By extending the existing UK vaccination program to all adolescents ≥12 years, our model has demonstrated a potential 21% reduction in hospitalization due to COVID, an 18% reduction in COVID-related deaths, and a 27% reduction in the prevalence of long COVID. If the vaccination schedule could be further extended to children ≥5 years, this benefit would be expanded to a 60% reduction in hospitalizations, a 57% reduction in mortality, and a 75% reduction in cases of long COVID. However, it is important to acknowledge that most of this direct benefit is not realized in adolescents and children receiving the vaccine, but largely in older individuals. 

England lifted its coronavirus restrictions on 19 July 2021, branded ‘Freedom Day’, despite a high community prevalence of SARS-CoV-2. More than 100 experts signed an open letter to the UK government suggesting that ending the pandemic requires enough of the population to be immune to prevent exponential growth of the virus, and population immunity is unlikely to be achieved without much higher levels of vaccination across all age-groups, and to include adolescents and children [13]. Our model supports the assertions of these experts, and demonstrates a significant, expected increase in the COVID-19 disease burden over the coming months as the population moves towards pre-pandemic levels of social mixing. However, despite the lifting of restrictions, reported cases in the UK fell in late July/early August. This observation has been postulated to be due to a variety of factors: the success of vaccines and NHS test-and-trace processes, a reduction in testing, a continuation of the vast majority of individuals maintaining social distancing and face coverings despite legislative changes, or the natural transmission break that occurred as adolescents and children entered summer vacation. For the two weeks prior to ‘Freedom Day’, up to 10% of adolescents and children were absent from school after confirmed COVID-19 contact [14]. This break in the viral transmission at schools among unvaccinated adolescents mimics our modelling of the impact from vaccinating this population, without adolescents and children missing their vital educational experience. Additionally, it highlights that, once children return to school, if a viral reservoir remains in the community, then there is likely to be an additional, larger wave of COVID-19 cases; this demonstrates the need to take a population-based approach to COVID-19 management. Our study demonstrates that this burden can be reduced by 35% if the current UK vaccination program is extended to all adolescents ≥12 years. These results align with those from a recent Australian modelling study, which evaluated the impact of SARS-CoV-2 vaccination under a range of coverage levels, vaccine efficacies, and age-groups, and which demonstrated that a high vaccination coverage of 90% in both adults and adolescents was required to mitigate COVID-19 transmission without implementing lockdown measures [15]. Furthermore, a study of SARS-CoV-2 vaccination in Israel provide observational evidence that, as well as protecting individuals who have been vaccinated, COVID-19 vaccination provided indirect protection to unvaccinated individuals; for every 20% point increase in the proportion of individuals who were vaccinated, the positive test fraction for the unvaccinated population decreased approximately two-fold [16].

Although COVID-19 illness is generally milder in children when compared with adults, with low risks of hospitalization and death [17,18,19], a small number of children present with a multisystem inflammatory state with an associated mortality rate of 1–2% [20,21,22]. Furthermore, evidence is emerging that children may experience long COVID symptoms, similar to adults, which can lead to high morbidity and limitations in daily functioning [23,24,25]. In adults, there is clear evidence of post-COVID damage to the lung, heart, kidneys, and brain. The ONS reported that 15% of young adults aged 16–29 years experience long COVID symptoms, with at least 35% reporting it affecting their work or education beyond 3 months [26]. It would be unlikely if this did not extend, to some degree, to lower age groups. Additionally, adolescents and children are affected by the indirect effects of COVID-19, for example, the interruption of schooling, the impact on mental health, increased social inequalities, and in a worst-case scenario, the loss of a parent or grandparent [27]. 

The benefits of vaccinating younger individuals are only relevant if they outweigh potential complications of vaccination. To date, the only significant adverse effect of mRNA vaccines being explored is the risk of myocarditis and pericarditis. There have been 1226 cases reported in around 300 million doses of either the Pfizer/BioNTech or Moderna vaccinations up to 11 June 2021, with a significant proportion occurring in adolescents (and predominantly male adolescents), more than any other age group [28]. However, there have been no recorded deaths due to myocarditis and pericarditis, and almost all cases have made a spontaneous, complete recovery. Even if we were to assume a doubling of risk in adolescents compared to adults >20 years (8 per million), extrapolation onto the adolescent population results in 37 excess cases in the UK (which would resolve spontaneously), compared to the several thousand hospitalizations and 23 deaths of adolescents prevented by vaccination. 

Currently there is no full authorization for children <12 years old, although the data from two mRNA vaccines have been used to grant provisional authorization for the Moderna and Pfizer vaccines [29,30]. Assuming these trials report no unexpected adverse events, our model predicts a greater population-wide benefit by extending the vaccination program to children than was observed in adolescents. Given the increased mixing patterns, the difficulties in maintaining physical distance, and the poor adherence to face coverings, this approach has double the impact of vaccinating those over the age of 12 (73% vs. 26%, respectively). This benefit translates into a 75% reduction in long COVID, a condition that we do not yet fully understand the longer-term impact of. 

### Limitations

This analysis assumes no further waning of vaccine efficacy over the duration of the study, whether this be time-dependent or based on further mutations of the virus. These assumptions, however, underestimate the potential impact of vaccinating the adolescent population given that the risk of severe infection in fully vaccinated adults will be greater. Similarly, the assumption that those who have previously been exposed and have recovered are no longer susceptible has recently been challenged, with a growing number of cases of the delta variant infecting people who have recovered from previous variants. Our model assumed that those of the UK population (approximately 10%) that have previously had confirmed COVID-19 were immune to further infection. This assumption, however, would underestimate the projections for cases and have little impact on the percentage reduction by vaccinating the adolescent population, given the very low incidence in this population. Further, we have not specifically modelled the impact on vulnerable populations. However, the data that were incorporated into the models did not stratify by vulnerability, but by age alone. Clearly, within each age range there will be sub-groups at greater risk, however this risk variability is a component of the data entered into the model and therefore can be reasonably applied to the outputs to determine those at personal, higher risk. Additionally, we modelled several different scenarios based around the population’s behavioral responses to the easing of mandated restrictions. When considering the impact of R_eff_ rising to 1.7 or 2.1, which both have precedents in western Europe, we see a robust percentage reduction in cases and hospitalizations, although the absolute numbers could be higher. Regarding hospitalizations, the UK model offers high-quality, centrally funded health care; outcomes may well be poorer in funding-constrained regions. Further, our conclusions may not extend to regions with very different social mixing patterns. Finally, we have not incorporated the seasonal effects on infection. Much of our knowledge regarding the delta variant has come from public health figures in the UK since April 2021. It is difficult to anticipate the impact of the autumn and winter on the virus. Again, this is likely to underestimate the potential impact of vaccinating adolescents, as indoor mixing, and therefore transmission, will be greater in winter.

## 5. Conclusions

The primary aim of the global vaccination program has always been to prevent COVID-related hospitalizations and deaths, and the data presented here clearly demonstrate that vaccinating adolescents and children has the potential to play a vital role in achieving this and in ending the pandemic more quickly. Indeed, a 60% reduction in hospitalizations and a 57% reduction in mortality could potentially bring the pandemic to an end without the need for further lockdowns or restrictions in movement. Not vaccinating adolescents and, when the technology is licensed, children, will make it substantially more difficult for the UK and other countries to reach the population immunity level needed to suppress the surging delta variant and reduce circulation of SARS-CoV-2.

## Figures and Tables

**Figure 1 vaccines-09-01180-f001:**
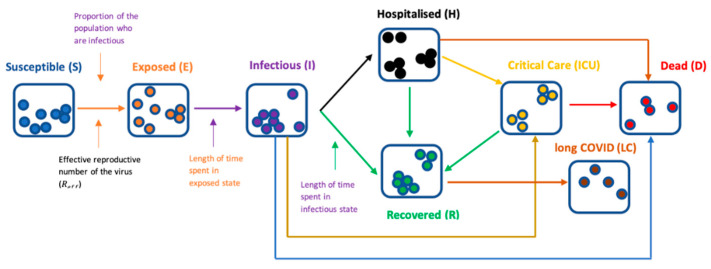
Flow diagram characterizing the transmission model.

**Figure 2 vaccines-09-01180-f002:**
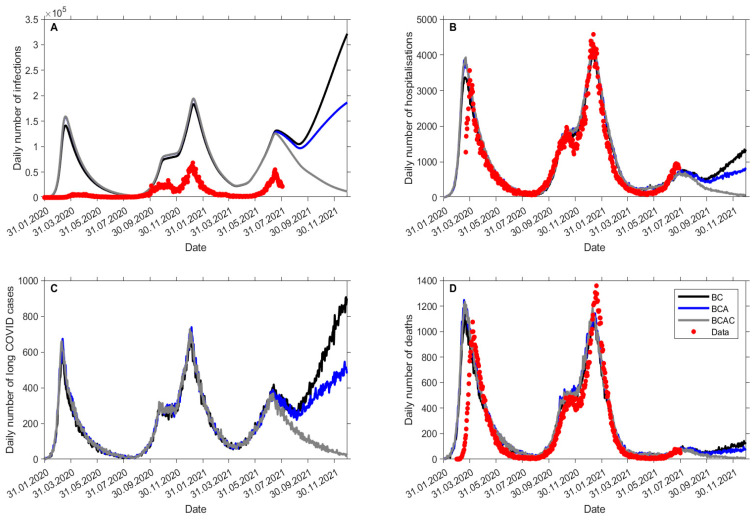
Expected total (**A**) SARS-CoV-2 infections, (**B**) COVID-19 hospitalizations, (**C**) long COVID, and (**D**) COVID-19 deaths between July and December 2021 by vaccination scenario. Note: projections in panels (**A**,**B**,**D**) are expected versus observed data.

**Figure 3 vaccines-09-01180-f003:**
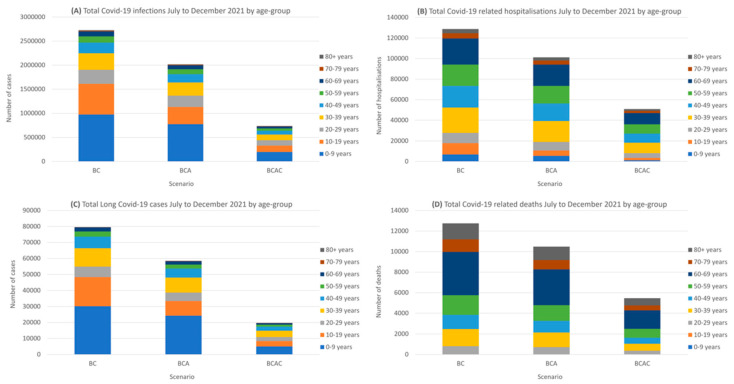
Total (**A**) SARS-CoV-2 infections, (**B**) COVID-19 hospitalizations, (**C**) long COVID, and (**D**) COVID-19 deaths expected between July and December 2021, stratified by vaccination scenario and age group.

**Figure 4 vaccines-09-01180-f004:**
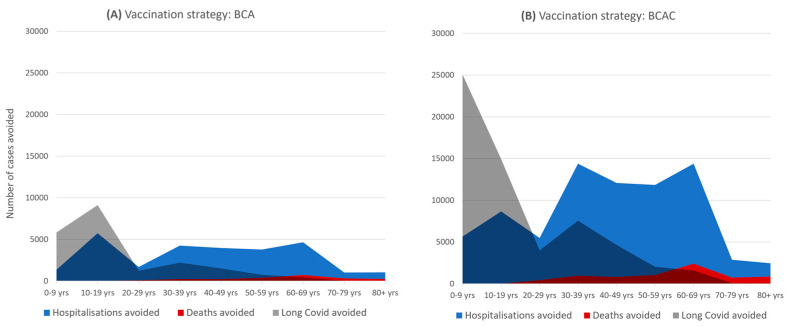
Expected events avoided (absolute numbers) between July and December 2021 by extending vaccination to (**A**) adolescents (BCA) and (**B**) adolescents and children (BCAC).

**Table 1 vaccines-09-01180-t001:** Summary of projected cumulative infections, hospitalizations, deaths, and long COVID cases in the UK between 19 July 2021 and 31 December 2021, for different scenario and sensitivity analyses.

Vaccination Strategy	Maximum Vaccine Coverage in Population	Effective Reproductive Number	Cumulative New Infections	Cumulative Hospitalizations	Cumulative Deaths	Cumulative Long COVID Cases
Base case (Adults 18+ years and vulnerable adolescents 12–17 years)	100%	**1.4**	**2,725,686**	**128,734**	**12,748**	**79,629**
		1.7	7,367,584	337,579	33,357	216,914
		2.1	19,094,941	1,081,038	113,907	554,046
	75%	1.4	2,989,312	139,418	13,628	88,164
		1.7	8,119,316	372,529	36,731	239,837
		2.1	19,648,560	1,121,021	117,717	571,990
	50%	1.4	3,118,892	144,440	14,085	92,901
		1.7	8,464,097	388,705	38,274	251,293
		2.1	19,744,737	1,128,337	117,742	578,284
Base case and adolescents (Adults 18+ years and adolescents 12–17 years)	100%	**1.4**	**2,016,731**	**101,210**	**10,488**	**58,484**
		1.7	5,552,442	260,481	26,793	161,184
		2.1	16,813,231	936,322	100,538	478,112
	75%	1.4	2,296,646	112,249	11,540	66,834
		1.7	6,162,484	285,963	28,792	179,285
		2.1	18,049,037	1,013,611	107,652	516,320
	50%	1.4	2,404,602	116,105	11,366	70,313
		1.7	6,962,520	319,802	31,928	205,390
		2.1	18,638,218	1,048,450	111,270	537,202
Base case and adolescents and children (Adults and children 5+ years)	100%	**1.4**	**735,910**	**50,928**	**5473**	**19,673**
		1.7	1,557,985	96,323	10,492	41,550
		2.1	7,590,762	429,352	47,730	201,420
	75%	1.4	896,826	57,348	6066	24,485
		1.7	2,228,161	124,566	13,296	60,579
		2.1	10,870,305	588,190	64,018	294,621
	50%	1.4	1,057,800	63,819	6702	28,865
		1.7	2,820,290	147,444	15,667	77,925
		2.1	12,871,903	686,218	73,961	353,840

Bold represents the main analysis, whereas the other rows are other scenarios explored in the sensitivity analysis.

## Data Availability

The datasets analyzed during the current study were sourced from and are available in the original publications referenced.

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
