# Peer review of "Vaccinating Adolescents and Children Significantly Reduces COVID-19 Morbidity and Mortality across All Ages: A Population-Based Modeling Study Using the UK as an Example"

_vaccines, 2021, doi:10.3390/vaccines9101180_

Round 1

Reviewer 1 Report

The paper provides a SEIR based mathematical methodology to predict the effect of SARS-CoV-2 vaccination in adolescents and children in the UK population. This is an important issue to access whether the vaccine programme of adolescents and children can reduce the COVID-19 disease burden, especially examining at different age groups. This manuscript outlines the including adolescents and children into the vaccine programme within high coverage rate will potentially reduce over 50% hospitalization and mortality of COVID-19 disease in the future. Other published papers have used similar SEIR mathematical methods to predict the case numbers of COVID-19 disease, although this report further stratified the population by age to analyze the potential benefit effect of vaccine programme among population <18 years.

Comments:

  1. Page 2, line 57-59: “Due to rare severe side-effects and deaths among those receiving the AstraZeneca vaccine, those aged under 30 are recommended to receive an mRNA vaccine from Pfizer/BioNTech or Moderna.” It is hard to understand that why only rare severe side-effects and deaths among those receiving the AstraZeneca vaccine contributes to the recommendation of Pfizer/BioNTech or Moderna for those age under 30.
  2. P. 4 line 141: “Assuming no change in vaccination strategy or social distancing measures, daily SARS-CoV-2 infections would approach approximately 300,000 per day.” Figure 2A showed the assuming numbers of infection; however, there are two black lines for the BC group in the figure 2A. In addition, if BC group was showed as the upper black line, the numbers of infection predicted by the model were approximate 3-folds of original UK data (red dots). In contrast to figure 2A, the numbers of covid-19 hospitalizations and deaths that were predicted by the model showed much close to UK data in figure 2B and 2D, respectively. Were the model settings or parameters fitted to the numbers of hospitalizations and/or deaths instead of the number of infection? More descriptions on the model development, such as the model settings or parameters, will aid to further understanding the results.
  1. Please define the long COVID in as well as Figure 2C, 3C, and other description through the manuscript. Is the definition followed the criteria from World Health Organization, UK government, or any references? In addition, the data in table S5 provided the proportion with long COVID (%) in different age groups. Did the results come from the clinicians? The footnote of table S5 showed that the data came from “Opinions and Lifestyle Survey” but without references or available website address.
  2. In the Table S9-S12 and Figure S1-S3, the proportion of individuals vaccinated were displayed in different age groups. Although the reference 15 was cited in the tables, the proportion in different age groups were not available in the cited website. Please recheck the website address or provide more specific references, especially for the data of proportions in adolescents and children groups. Also, Table S10-S12 showed the forecasted age dependent vaccination coverage overtime. How were they forecasted? Please add the descriptions in the method or supplementary section.
  3. Please provide the website addresses or other references citations for the reference 5, 6, 7, 8, 11, 24, 25.
  4. The volume and page number were lost for reference 10 and 13.

Author Response

Reviewer 1

  1. Page 2, line 57-59: “Due to rare severe side-effects and deaths among those receiving the AstraZeneca vaccine, those aged under 30 are recommended to receive an mRNA vaccine from Pfizer/BioNTech or Moderna.” It is hard to understand that why only rare severe side-effects and deaths among those receiving the AstraZeneca vaccine contributes to the recommendation of Pfizer/BioNTech or Moderna for those age under 30.

Author response

To aid clarity, this section of paper has been edited and now reads:

‘Currently, the AstraZeneca (Oxford), BioNTech/Pfizer and Moderna Covid-19 vaccines are available across the UK, all of which provide effective protection against coronavirus infection, and even better protection against serious illness and disease. There have been a small number of cases of severe side-effects and deaths among those receiving the AstraZeneca vaccine. However, for people aged 30 or over and those with other health conditions, the benefits of being vaccinated outweigh any risk associated with this vaccine. For people under 30 without other health conditions, it is advised that it they receive an mRNA vaccine from Pfizer/BioNTech or Moderna as an alternative to this vaccine.’

  1. P. 4 line 141: “Assuming no change in vaccination strategy or social distancing measures, daily SARS-CoV-2 infections would approach approximately 300,000 per day.” Figure 2A showed the assuming numbers of infection; however, there are two black lines for the BC group in the figure 2A. In addition, if BC group was showed as the upper black line, the numbers of infection predicted by the model were approximate 3-folds of original UK data (red dots). In contrast to figure 2A, the numbers of covid-19 hospitalizations and deaths that were predicted by the model showed much close to UK data in figure 2B and 2D, respectively. Were the model settings or parameters fitted to the numbers of hospitalizations and/or deaths instead of the number of infection? More descriptions on the model development, such as the model settings or parameters, will aid to further understanding the results.

Author response

Whilst it may appear as if there are two black lines for the BC group in the figure 2A, this is not the case and what appears as the lower black line is in fact the outer margin of the red dots. To avoid confusion, we have now removed this outer margin in the figure.

Due to word constraints for the manuscript, further description on the model development are provided in the supplementary material. To clarify, the model was not optimised to the observed data but instead the model inputs were used to re-produce the outcomes. The observed cases reported underestimates the number of people infected as there is likely to be a large proportion of the population who are infected but have mild or no illness, and thus do not require medical care and do not undergo testing. However, these individuals may still contribute to virus transmission in the population.

  1. Please define the long COVID in as well as Figure 2C, 3C, and other description through the manuscript. Is the definition followed the criteria from World Health Organization, UK government, or any references? In addition, the data in table S5 provided the proportion with long COVID (%) in different age groups. Did the results come from the clinicians? The footnote of table S5 showed that the data came from “Opinions and Lifestyle Survey” but without references or available website address.

Author response

We have defined long Covid (and referenced the WHO’s briefing on Long Covid) after first use of the term in the introduction:

‘It has estimated that at least one in five people infected with Covid-19 experience persisting ill health following the acute phase of infection. This condition is typically referred to as long Covid, and includes both ongoing symptomatic Covid-19, with signs and symptoms continuing for 4 to 12 weeks after acute Covid-19, as well as post-Covid-19 syndrome, with signs and symptoms that develop during or after acute Covid-19, continuing for more than 12 weeks, and which cannot be explained by an alternative diagnosis[2].’

We apologise for our omission of the reference for the proportion with long COVID (%) in different age groups. This reference has now been added:

Office for National Statistics. Coronavirus and the social impacts of ‘long COVID’ on people’s lives in Great Britain. 2021. Available from:

https://www.ons.gov.uk/peoplepopulationandcommunity/healthandsocialcare/conditionsanddiseases/articles/coronavirusandthesocialimpactsoflongcovidonpeopleslivesingreatbritain/7aprilto13june2021.

  1. In the Table S9-S12 and Figure S1-S3, the proportion of individuals vaccinated were displayed in different age groups. Although the reference 15 was cited in the tables, the proportion in different age groups were not available in the cited website. Please recheck the website address or provide more specific references, especially for the data of proportions in adolescents and children groups. Also, Table S10-S12 showed the forecasted age dependent vaccination coverage overtime. How were they forecasted? Please add the descriptions in the method or supplementary section.

Author response

Ref 15 was incorrectly added to tables S10-S12 as these are forecasted vaccination coverage.

Table S9 contains previous vaccine coverage and the ref is applied here. However, we apologise as the ref that was inserted was incorrect. We have now updated the correct ref in which the vaccination coverage by age group can be found: Statistics » COVID-19 Vaccinations (england.nhs.uk). We used the weekly data which is reported in various excel spreadsheets. On opening the excel data sheets, there is a worksheet titled ‘ICS STP’ with % who have had at least 1 dose (using ONS denominators) and those are the numbers we used for our model.

With regards to the forecasted age dependent vaccination coverage, we used the observed rate of vaccination to forecast future coverage speed. The number of people who were vaccinated between 18 July and 25 July, were assumed to have been vaccinated weekly going forward until maximum coverage for that age group. We have added this information to the supplementary appendix on page 4.

  1. Please provide the website addresses or other references citations for the reference 5, 6, 7, 8, 11, 24, 25.

Author response

Website addresses have now been inserted for these references.

  1. The volume and page number were lost for reference 10 and 13.

Author response

Volume and page numbers have been inserted for these references.

Reviewer 2 Report

In this study, the authors developed a novel SEIR mathematical disease transmission model that quantifies the impact of different vaccination strategies on population-level SARS-CoV-2 infections and clinical outcomes,  using the UK as an example. 

The paper can be accepted for the publication after some major revisions.

  1. Use the present time in the abstract. For example, change ”We developed”to ”We develop”.
  2. The authors wrote they developed a mathematical model, but they just presented the schematic diagram of this model. So, the mathematical equations which represent this model should be added in the manuscript.
  3. The introduction section should be improved by adding some references about the mathematical modeling and its role in epidemiology.
  4. To help the reader, add a definition for effective reproductive number and its biological meaning.
  5.  What the difference between the effective reproductive number and the basic reproduction number?
  6. Compare your model and results with others existing in the literature.
  7.  There are some typos. The authors should carefully read the manuscript.
  8. When a disease spreads within a community, individuals acquire knowledge about this disease. It will be interesting to study the memory effect on the spreading of Covid-19 in presence of vaccination by using the new generalized fractional derivative presented in [1].
  9. Include and comment in the manuscript the recent and related works:
    [1] A new generalized definition of fractional derivative with nonsingular kernel, Computation 8 (2020) 1-9.                                        [2] A  delayed SIR epidemic model with general incidence rate, Electronic Journal of Qualitative Theory of Differential Equations 3 (2013) 1-9.

Author Response

Reviewer 2

  1. Use the present time in the abstract. For example, change ”We developed” to ”We develop”.

Author response

This sentence in the abstract has been changed to 'In this study, we present a novel SEIR mathematical disease transmission model'.

  1. The authors wrote they developed a mathematical model, but they just presented the schematic diagram of this model. So, the mathematical equations which represent this model should be added in the manuscript.

Author response

Due to constraints on word limits within the main body of the manuscript, the equations for the model can be found on pages 2 and 3 of the supplementary material.

  1. The introduction section should be improved by adding some references about the mathematical modeling and its role in epidemiology.

Author response

We have expanded the section around mathematical modelling in the introduction and added relevant references where appropriate. The sections now reads:

‘In recent years our understanding of infectious-disease epidemiology has been greatly increased through mathematical models, which have generated quantitative information to inform public health planning and infectious disease control[9-11]. Such mathematical models have become a central tool to control Covid-19 in many countries, whereby they can provide short- and long-term forecasts regarding the future direction of the Covid-19 pandemic and provide a framework for evaluating the potential impact of different vaccination strategies. The most relevant mathematical models relating to the spread of a pandemic are compartmental models, with the population divided into different population groups based on their infection status (e.g., susceptible, exposed, infectious, recovered) and with assumptions being made about the rate of transfer from one group to another.’

  1. To help the reader, add a definition for effective reproductive number and its biological meaning.

Author response

We have added the following at the first mention of effective reproductive number:

‘defined as the number of secondary infections produced by a single infectious individual when the population has both susceptible and non-susceptible (e.g., infectious, immune, vaccinated) individuals and/or interventions are in place to combat the spread of infection’.

  1. What the difference between the effective reproductive number and the basic reproduction number?

Author response

The basic reproduction number is the expected number of new infections caused by a single infectious individual in a homogeneous population where everyone is susceptible, i.e., before widespread immunity starts to develop and before a vaccination programme has been introduced.

The effective reproductive number is the expected number of new infections caused by an infectious individual in a population where some individuals may no longer be susceptible.

  1. Compare your model and results with others existing in the literature.

Author response

There are limited published modelling studies examining the impact of a vaccination strategy that includes <18 year olds, on the Covid pandemic. We have added the following to the manuscript:

‘These results align with those from a recent Australian modelling study which evaluated the impact of SARS-Cov-2 vaccination under a range of coverage levels, vaccine efficacies, and age-groups, and which demonstrated that a high vaccination coverage of 90% in both adults and adolescents was required to mitigate Covid-19 transmission without implementing lockdown measures[15]. Furthermore, a study of SARS-CoV-2 vaccination in Israel provide observational evidence that as well as protecting individuals who have been vaccinated, Covid-19 vaccination also provided indirect protection to unvaccinated individuals; for every 20% point increase in the proportion of individuals who were vaccinated, the positive test fraction for the unvaccinated population decreased approximately two-fold[16].’

  1. There are some typos. The authors should carefully read the manuscript.

Author response

The manuscript has been proof-read, and typos corrected.

  1. When a disease spreads within a community, individuals acquire knowledge about this disease. It will be interesting to study the memory effect on the spreading of Covid-19 in presence of vaccination by using the new generalized fractional derivative presented in [1].

Author response

We thank the reviewer for the suggestion. Whilst we agree that it will be interesting to study the memory effect on the spreading of Covid-19 in presence of vaccination, this is out of scope of our study. Therefore, we have not added the suggested reference to the manuscript at this stage, but would welcome any specific guidance as to how it could be incorporated.

  1. Include and comment in the manuscript the recent and related works:

[1] A new generalized definition of fractional derivative with nonsingular kernel, Computation 8 (2020) 1-9.                                        [2] A  delayed SIR epidemic model with general incidence rate, Electronic Journal of Qualitative Theory of Differential Equations 3 (2013) 1-9.

Author response

We thank the reviewer for the suggested references. Unfortunately, on close reading we could not determine exactly how these references align with the scope of our study. Therefore, we have not added them to the manuscript at this stage, but would welcome any specific guidance as to how they could be incorporated.

Round 2

Reviewer 2 Report

Now, the paper is well written and well organized.  Therefore, I recommend it for the publication in the journal.